

# Lasing in optically induced gap states in photonic graphene

**Marijana Milićević[1], Olivier Bleu[2], Dmitry D. Solnyshkov[2], Isabelle Sagnes[1],
Aristide Lemaître[1], Luc Le Gratiet[1], Abdelmounaim Harouri[1] Jacqueline Bloch[1],
Guillaume Malpuech[2] and Alberto Amo[3⋆]**

**1** Centre de Nanosciences et de Nanotechnologies (C2N),
CNRS - Université Paris-Sud/Paris-Saclay, Palaiseau, France
**2** Institut Pascal, CNRS/University Clermont Auvergne,
4 avenue Blaise Pascal, 63178 Aubiere, France
**3** Univ. Lille, CNRS, UMR 8523 – PhLAM – Physique des
Lasers Atomes et Molécules, F-59000 Lille, France

⋆ alberto.amo-garcia@univ-lille.fr

## Abstract

We report polariton lasing in localised gap states in a honeycomb lattice of coupled micropillars. Localisation of the modes is induced by the optical potential created by the excitation beam, requiring no additional engineering of the otherwise homogeneous polariton lattice. The spatial shape of the gap states arises from the interplay of the orbital angular momentum eigenmodes of the cylindrical potential created by the excitation beam and the hexagonal symmetry of the underlying lattice. Our results provide insights into the engineering of defect states in two-dimensional lattices.

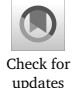
# 1 Introduction

The recent implementation of two-dimensional lattices of coupled photonic resonators provides a new sandbox for the study of propagation and localisation of photons with engineered dispersions [1–6]. By selecting the geometry of the lattice it has been possible to implement photonic dispersions that mimic those of electrons in certain solid state materials like graphene [6,7] and the polyacetilene macromolecule [8–11]. Beyond lattice geometries that already exist in nature, the control of the onsite energy and hopping between resonators has allowed the engineering of synthetic two-dimensional photonic lattices with exotic dispersions. Two significant examples are lattices with flat [12–14] and fractal bands [15, 16], in which the effects of localisation can be studied with exquisite precision. Up to now, most works in this domain have focused on the understanding of the bulk properties, particularly, the lasing dynamics [2, 3, 12–14] and the topological properties [17, 18]. Similar studies have been performed in passive lattices of waveguides, which have provided valuable information on the effects of bulk localisation [19–21] and propagation in topological edge states [22, 23].

An interesting feature of photonic lattices is the possibility of engineering states in the photonic gaps. They provide localised modes isolated in energy from the photonic bands. Their properties can eventually be tailored to create, for instance, localised lasing modes [5, 24], or used as precursors of lattice solitons [25–29]. A method to create isolated gap states is to implement one-dimensional lattices with nontrivial topology. At their edges, or when two lattices with different topological invariants are interfaced, localised states whose energy lies in the middle of the gap appear [30, 31]. Lasing in this kind of topological modes has been recently predicted [8, 32] and observed in one-dimensional lattices of micropillars [9] and of semiconductor ring resonators implementing the driven-dissipative version of the Su-Schrieffer-Heeger Hamiltonian [10, 11].

In the bulk of a homogeneous lattice, the only way to engineer a localised mode is to locally break the lattice periodicity. This can be done, for instance, via a local variation of the on-site potential. The so-called defect states emerge from the top or bottom of the photonic bands and get deeper into the gaps when the local perturbation is increased. This idea has been exploited to create localised photonic modes in subwavelength photonic crystal slabs [33].

A very suitable system to study the emergence of defect modes in lattices of resonators are semiconductor microcavities. Their eigenstates are polaritons, mixed light matter quasiparticles arising from the strong coupling between quantum well excitons and photons confined in a microcavity [34]. One of the main advantages of this system is the fact that the local potential can be varied with an external optical beam: nonresonant excitation of the microcavity injects a reservoir of excitons that interact repulsively with polaritons. In this way, the reservoir induces a local blueshift that can be used to shape the local potential [35–38]. This feature was employed in Refs. [5, 24] to demonstrate the emergence of localised gap states in a one-dimensional microcavity wire subject to a periodic photonic potential. In those works, it was shown that the symmetric or anti-symmetric character of the defect state depends on the specific point in the periodic potential at which the periodicity is broken.

In two-dimensional photonic lattices, defect modes have been hardly explored in a controlled way. Yet, they play an important role in the transport properties of graphene [39], and are at the origin of antibunched emission in $WSe_2$ and hBN [40, 41]; their study in photonic lattice simulators can provide insights into their properties. In this work, we study the appearance of defect gap states in a honeycomb lattice of coupled polariton micropillars. In the absence of a local perturbation, the lattice shows two sets of bands separated by a gap. Each set of bands arises from the coupling of either s- or p-modes confined in each micropillar [7]. When optically modifying the on-site potential, localised states emerge from the top of the highest s-band. If pumping is localised on a single pillar of the lattice, we observe lasing from

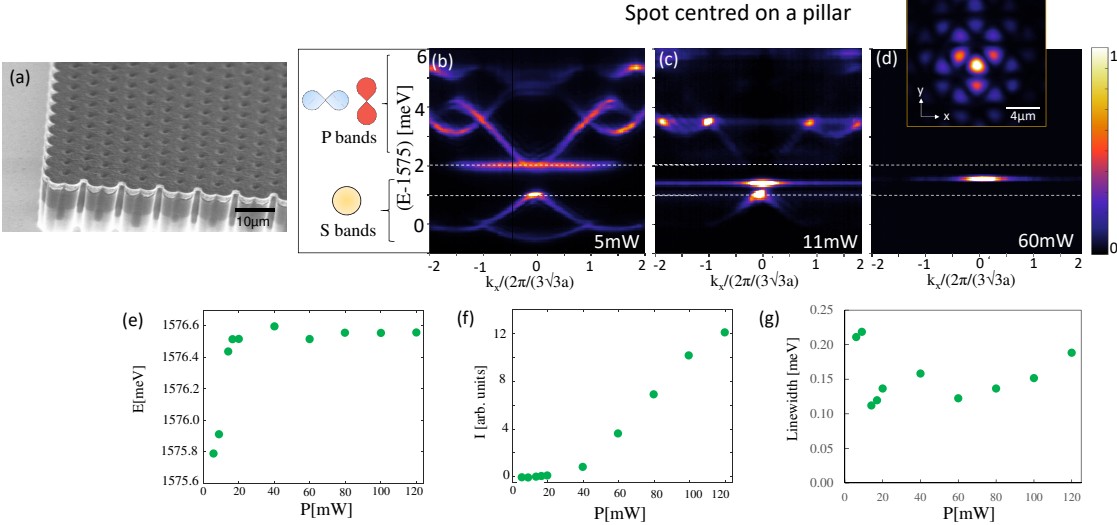

Figure 1: (a) Scanning electron microscope image of the polariton honeycomb lattice under study. (b) Energy and in-plane momentum resolved emission from the centre of the lattice under low excitation power (5 mW). The emission is recorded for $k_y = 0.87\ \mu\text{m}^{-1}$, for an excitation spot centred on a single pillar. The low power emission is virtually identical for a spot centred in a hole of the lattice. Dashed lines show the lower and upper limits of the gap between s- and p-bands. (c)-(d) Emission at 11 mW and 60 mW, showing the emergence of a localised gap state. The inset in (d) shows the real space emission of the gap state at 60 mW. (e)-(f) Energy and emission intensity of the gap state measured as a function of excitation power. (g) Measured linewidth (full width at half maximum) of the gap state extracted from a Gaussian fit to its spectral profile.

a single gap state above a certain pumping threshold, whereas if the pump is centred in the middle of a hexagon, lasing takes place simultaneously in three ring-shaped modes containing different angular momentum components. Our results clarify the picture of the emergence of gap states from band modes in a hexagonal lattice, and show that in a polariton lattice, a local excitation is enough to create tightly confined lasing modes.

## 2 The polariton honeycomb lattice

In the present experimental study we use a honeycomb lattice of coupled micropillars similar to that of Ref. [7]. The lattice, shown in Fig. 1(a), is chemically etched from a $Ga_{0.05}Al_{0.95}As$ $\lambda/2$ planar microcavity surrounded by two $Ga_{0.05}Al_{0.95}As/Ga_{0.8}Al_{0.2}As$ Bragg mirrors with 28 (upper mirror) and 40 (lower mirror) pairs grown by molecular beam epitaxy. The microcavity contains three sets of four GaAs quantum wells of 7 nm in width, located at the three central maxima of the electromagnetic field confined in the cavity. Strong light matter coupling between quantum well excitons and confined photonic modes results in a Rabi splitting of 15 meV. The experiments are performed at 4 K at exciton-photon detunings between -12 and -17 meV. The micropillars forming the lattice are 2.7 $\mu$m in diameter and their center to center distance $a$ is 2.4 $\mu$m. The sample is excited with a continuous wave laser at 740 nm, about 100 meV above the lower polariton emission energy. To avoid thermal effects, we modulate the laser with an acousto-optic modulator at 0.8 kHz with square gates and a duty cycle of 0.08%. The large numerical aperture of the excitation/collection microscope objective results

in a pump spot of $w = 3$ $\mu$m full width at half maximum. Photoluminescence from the sample is collected by a spectrometer coupled to a CCD with which we measure the spatial and momentum distribution of the emission as a function of energy.

Figure 1(b) shows the low power (5 mW) photoluminescence of the lowest energy bands in momentum space, with an excitation spot centred on one of the micropillars of the lattice. To avoid destructive interference effects characteristic of the emission from the first Brillouin zone [7], we collect the light emitted through the $K - \Gamma - K'$ line in momentum space centred on the second Brillouin zone ($k_y = 0.87$ $\mu$m$^{-1}$). The lowest energy bands arise from the coupling of the lowest polariton mode in each micropillar, of cylindrical symmetry. These bands present a dispersion analogous to that of electrons in graphene. A fit of the measured dispersion to a tight-binding model gives a hopping amplitude of $t = 0.26$ meV [7]. The asymmetry between upper and lower s-bands can be described within the tight-binding model by a phenomenological next-nearest neighbours hopping of $t' = -0.03$ meV. A gap, marked by dashed lines in Fig. 1(b)-(d), separates these bands from the four p-bands that arise from the coupling of the first excited modes in the micropillars. From tight-binding fits we estimate a nearest-neighbour hopping amplitude of $-1.2$ meV for p-orbitals oriented parallel to the links between micropillars, and 0 for the hopping of polaritons in orbitals oriented perpendicular to the links (see Refs. [7, 18]).

## 3   Excitation centred on a micropillar

When the excitation density is increased, a state detaches from the top of the s-bands entering the gap between the s- and p-bands (Fig. 1(c)-(e)). The energy of this gap state increases when rising the excitation power (Fig. 1(e)), and above a threshold of 40 mW, lasing on this mode is triggered. This is attested by the nonlinear threshold in the emitted intensity as a function of the excitation density (Fig. 1(f)), and the simultaneous linewidth reduction by a factor of 2 across the threshold (Fig. 1(g)). Above the lasing threshold power, nonlinear effects arising from the excited charges in the quantum wells induce spectral wandering of the gap state and degrade the linewidth in our time-integrated measurement.

We can understand the origin of this gap state from the potential locally induced by the polariton reservoir injected by the excitation laser. When exciting the microcavity non-resonantly, that is, at an energy exceeding the quantum well band gap, hot electrons and holes are injected in the microcavity. They relax down to form a reservoir of polaritons that accumulate at high momenta close to the exciton energy, 20 meV above the s-bands. From there, polaritons relax down to the lower polariton bands. The polariton reservoir interacts repulsively with the lowest band polaritons inducing a blueshift. Due to the heavy mass of the reservoir states (close to that of free excitons i.e. $\sim 0.9$ $m_e$, with $m_e$ being the free electron mass) the diffusion length of the reservoir is very short, of the order of a few microns [36]. The induced potential is thus localised under the laser excitation spot, in this case centred on a micropillar of the lattice. The reservoir potential creates a localised state that penetrates into the gap (defect state), very similarly to what was observed in a 1D microcavity with a periodic modulation [5]. Electro-optics effects can be discarded to be at the origin of the local blueshift due to the very low cw powers used in our experiments compared to the high peak energies required [42].

The hexagonal symmetry of the lattice determines the shape of the gap state, which shows a three points star shape, as depicted in the inset of Fig. 1(d) for an excitation density of 60 mW, above the lasing threshold. At this power, the emission is fully dominated by the gap state, and the real space image in Fig. 1(d) is recorded using a band-pass filter that eliminates the excitation laser light. The zero of intensity between the lobes witnesses the anti-symmetric nature of the state, with a phase change of $\pi$ between adjacent lobes of the wavefunction. The

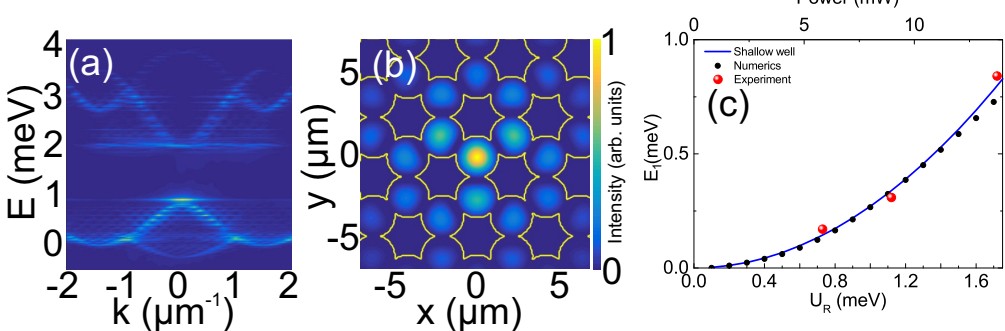

Figure 2: (a) Calculated momentum space dispersion of a lattice with an external potential localised in a single pillar with an amplitude $U_R = 0.5$ mW. s- and p-bands as well as a gap state close to the top s-band are visible. (b) Calculated real-space emission intensity of the gap state. (c) Black dots: Calculated energy of the gap state relative to the top of the s-band as a function of the reservoir potential (lower scale). Solid line: Eq. 2. Red dots: measured energy of the gap state from the top of the s-band as a function of the excitation density (upper scale) for low excitation densities ($< 18$ mW, Fig. 1(e)). The relative extent of the upper horizontal scale has been chosen to overlap the experimental points to the calculation.

anti-symmetric character of the state is inherited from the shape of the wavefunctions of the anti-bonding top s-band [7].

The significant localisation of the gap state explains why lasing is favoured in this state versus the bulk modes. Its localisation in an area comparable to the excitation laser spot optimizes the overlap of this state with the reservoir, resulting in a lower threshold for the gap state than for the bulk, propagating modes.

To provide a qualitative description of the spatial distribution and energy of the gap states observed in the experiment, we solve the time-independent 2D Schrödinger equation describing the cavity field $\psi(x, y)$ in the absence of pump and losses:

$$E\psi(x, y) = -\frac{\hbar^2}{2m}\Delta\psi(x, y) + (U(x, y) + U_{res}(x, y))\psi(x, y), \qquad (1)$$

where $m$ is the effective mass of the lower polariton branch at $k = 0$ in the unetched planar microcavity ($m = 4.9 \cdot 10^{-5} m_e$). $U(x, y)$ is the confinement potential of polaritons with a value of 0 in the region where the micropillars are present, and 20 meV in the etched away region (the barrier height is a fitting parameter, chosen to reproduce the dispersion, especially the gap size). $U_{res}(x, y)$ is the potential created by the reservoir polaritons, which we take as a Gaussian with amplitude $U_R$ and width $w = 3$ $\mu$m (the size of the laser spot in the experiment) centred on a micropillar. The solution of this equation on a grid gives the eigenenergies $E$ and the eigenstates of the s- and p-bands separated by a gap. When the on-site Gaussian potential $U_R$ is increased, the simulation shows the emergence of a state from the top of the s-band which enters the gap, as can be seen in Fig. 2(a) plotted in momentum space. The real-space shape of this state for an amplitude $U_R = 0.5$ meV is shown in Fig. 2(b) (yellow lines show the confinement potential $U$). The analysis of the phase of the calculated wavefunction of the mode confirms the anti-symmetric nature of the gap state.

The calculated energy of the gap state created by the reservoir is shown in solid black dots in Fig. 2(c). Its apparent quadratic dependence as a function of $U_R$ can be described analytically using a simple model of a shallow potential well [43]. Polaritons on top of the upper s-band behave like quasi-particles with negative effective mass. Due to the negative sign

of the mass, a positive localised potential $U_R$ results in the trapping of polaritons in a localised state, with an energy higher than that of the top of the s-band. For a shallow potential well of "depth" $U_R$ and size $w$ (where shallow means $U_R \ll \hbar^2/2|m^*|w^2$, which can be fulfilled either by reducing $U_R$, $w$, or both), the approximate energy of the first confined state (positive, measured from the top of the s-band) takes the form

$$E_1 \approx \frac{|m^*|w^2}{2\hbar^2}U_R^2, \tag{2}$$

where $m^*$ is the (negative) effective mass at the top of the band, which we obtain from the experimentally determined hopping coefficients $t$ and $t'$ (see above) and a parabolic band approximation around the $\Gamma$ point $(k_x, k_y) = (0,0)$. This energy is plotted in Fig. 2(c) as a blue line, which shows a good agreement both with the numerical simulations, and with the first three experimental points of Fig. 1(e), plotted here as red dots. These three points are measured below the lasing threshold, where the reservoir potential $U_R$ is expected to increase linearly with the pumping (above threshold, the stimulated scattering process results in a sublinear increase of the reservoir density). In Fig. 2(c) the horizontal scale for the red dots is the excitation power (upper scale). Its span has been adjusted to match the calculated energy dependence of the gap state as a function of $U_R$.

## 4 Lasing from gap modes induced by a hexagonal potential

The flexibility in shaping the local potential with the use of the injected reservoir allows exploring other gap states, in particular, states involving angular momentum modes. This is the case when the pump is centred on a hole of the hexagonal lattice. As the pump spot is wider than the size of the hole ($\sim 1~\mu$m), the injected reservoir induces a local blueshift on the six micropillars encircling the hole. Since the induced repulsive potential is more extended than in the case discussed above of pump centred on a single pillar, the shallow well picture does not apply any more and several localized gap modes are expected.

Figure 3 shows the measured momentum space emission as a function of the excitation power for such hole-centred spot. At 25 mW we observe a set of states detaching from the upper s-band and entering the gap. At 35 mW, up to six gap states can be identified. When the power is increased above 45 mW, lasing takes place simultaneously in three of them, labelled S1, S2, and S3, up to the highest available power in our set-up. The lasing threshold is characterised by a nonlinear increase of the emitted intensity with excitation density (Fig. 3(g)) and by an abrupt linewidth reduction (see Fig. 3(h) for state S1 – similar features are observed for states S2 and S3).

Insights into the nature of the three gap states can be revealed from their real- and momentum-space distributions at an excitation power of 105 mW. Figure 4(b)-(d) shows the spatial distribution of the gap modes. In this measurement, the emitted intensity is filtered in a spectrometer with a narrow slit (spectral resolution of 60 $\mu$eV) attached to a CCD camera. By recording the spectra for different positions of the imaging lens in front of the entrance slit, we can reconstruct the real space pattern for any emission energy [9,44], in this case the energies of the S1, S2, and S3 modes. As a reference, Fig. 4(a) shows the bulk extended emission from the top of the upper s-band at 15 mW, below the excitation power required to create the gap states. The extension of the emission far from the excitation spot (located at the central hole of the image) proves its bulk nature, while the radial decay of the emission arises from the cavity radiative losses. In contrast, at high excitation power, gap state S1 is highly localised in the central hexagon (Fig. 4(b)). The emission pattern shows a clear $C_{6v}$ symmetry following that of the underlying hexagonal lattice, with zeros of intensity in between the micropillars.

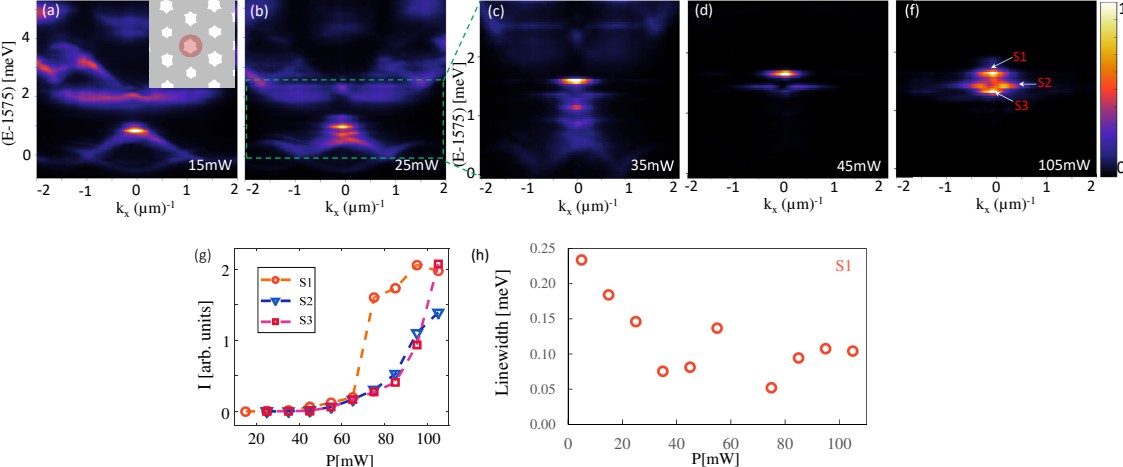

Figure 3: (a)-(f) Momentum-space emission as a function of power for an excitation spot centred on one of the hexagons of the lattice, as sketched with a red circle in the inset of (a). For each image, the linear colour scale has been normalised to the maximum intensity. (g) Measured emission intensity as a function of excitation power for the three lasing gap modes. (h) Measured linewidth (full width at half maximum) of the S1 gap state extracted from a Gaussian fit to its spectral profile.

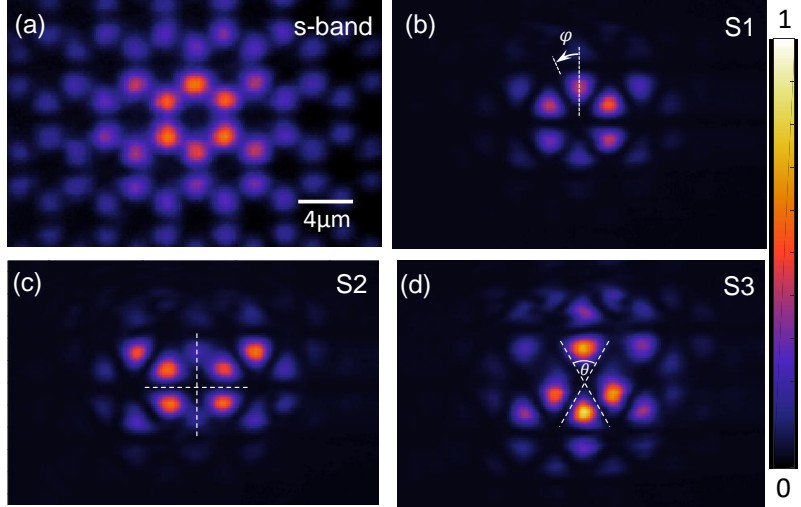

Figure 4: Real-space of emission of the energy selected top states of the s-band at low power 15 mW (a), and gap modes S1 (b), S2 (c) and S3 (d) at 105 mW. The intensity in each panel has been independently normalised.

The emission pattern for gap states S2 and S3 is quite different. S2 shows four intense quasi-circular spots in the central hexagon with perpendicular separatrix, while in S3 two of the four central spots are elongated and the separatrix present an angle $\theta \sim 60°$. For S2 and S3, away from the four central spots, the emission recovers a hexagonal pattern. Note that the emission patterns shown in Fig. 4 and in all the other figures in this paper are measured with a linearly polarised filter oriented in the horizontal direction with respect to Fig. 4. All other polarisations show the same patterns; any eventual splitting between states of different polarization is smaller than the linewidth of these states and, therefore, if present, it could not be measured.

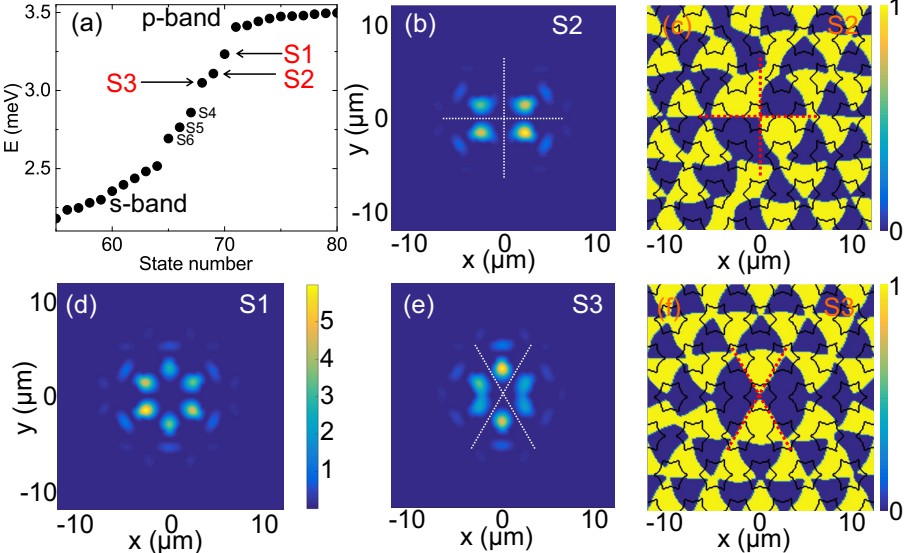

Figure 5: (a) Calculated eigenvalues of the honeycomb lattice subject to a local potential of 3 meV in the central hexagonal cluster. Only the region around the gap between s- and p-bands is shown. (b), (d), (e) Calculated squared absolute values of the eigenfunctions corresponding to the S2, S1, and S3 gap states in horizontal polarisation (identical shapes are found for the vertically polarised modes). (c), (f) Phase of the eigenfunctions corresponding to S2 and S3. The phase is displayed in units of $\pi$.

The symmetry of the real space pattern of states S2 and S3 is $C_{2v}$, i.e., invariant under rotations of 180°. To understand this symmetry reduction with respect to the $C_{6v}$ symmetry of the hexagonal lattice, we have to analyse the properties of the eigenfunctions of such gap states. This analysis is performed in the next sub-sections first by solving numerically the full 2D Schrödinger equation (Eq. 1) with a Gaussian reservoir centred in a hexagon of the lattice. We then propose an analytical approach, based on the envelope function approximation which explains qualitatively the peculiar shape of states S2 and S3. We discuss the difference between these states and the eigenstates of a bare photonic benzene molecule [44].

## 4.1 Numerical simulations

The results of the numerical simulation based on Eq. 1 for a pump spot centred in a hole of the lattice are presented in Fig. 5. Panel (a) shows the energies of the states close to the gap as a function of the eigenvalue number (related to the total number of pillars in the considered structure), calculated for an amplitude of the reservoir potential $U_R = 3$ meV. Six modes S1-S6 appear in the gap. The states of interest here, S1-S3, are located in its upper part. States S2 and S3 would be degenerate in a perfectly symmetric system. In our simulations, in order to reproduce the splitting observed in the experiment, we lift the degeneracy by introducing a 10% ellipticity in the shape of the pump, with the long axis in the $x$ direction. We have checked that the spatial gradient due to the inhomogeneous cavity thickness present in our sample (of about 6 $\mu$eV/$\mu$m) is not enough to induce the large measured splitting of 60 $\mu$eV between S2 and S3. The calculated spatial distributions of the intensity and phase of states S1-S3 are shown in Fig. 5(b)-(f). The order in energy of the simulated states, the number of lobes, and their orientation correspond to the experimental findings. Therefore, the pump ellipticity seems to be at the origin of the observed symmetry breaking: the degenerate states S2+S3 conserve the $C_{6v}$ symmetry, whereas each of them alone (after degeneracy lifting)

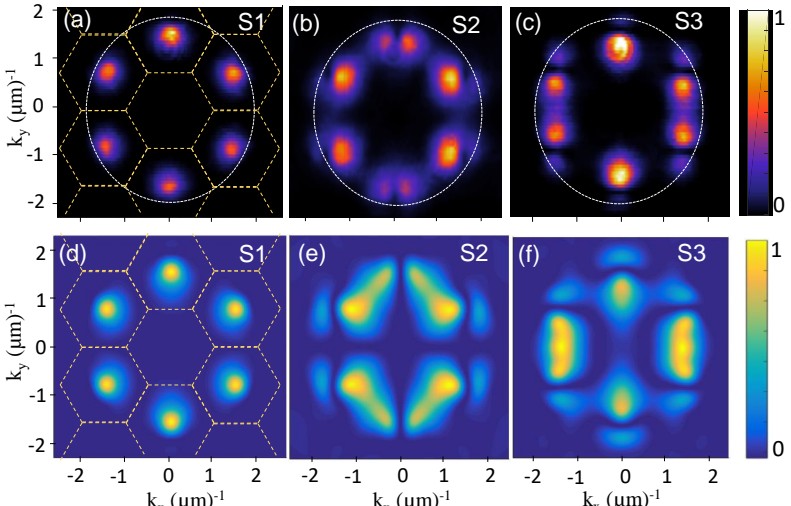

Figure 6: (a)-(c) Measured momentum space patterns of the S1, S2 and S3 gap states. Hexagons in (a) represent the reciprocal Bravais lattice. Dotted circles indicate the numerical aperture of the imaging setup. The intensity in each panel has been independently normalised. (d)-(f) Momentum space of the gap states obtained from direct Fourier transform of the calculated eigenfunctions shown in Fig. 5.

possesses only a $C_{2v}$ symmetry (with the orientation determined by the elongated shape of the laser spot). State S2 shows orthogonal separatrix between lobes (Fig. 5(b),(c)), while state S3 (Fig. 5(e),(f)) presents separatrix crossing at 60° and 120°, as in the experiment.

The measured and calculated Fourier images of the gap states S1-S3 are presented in Fig. 6. The dotted lines in panels (a)-(c) show the numerical aperture of the experimental setup. Good agreement between experiment (panels (a)-(c)) and simulations (panels (d)-(f)) for all states confirms the validity of our model. The Fourier transform conserves the number of lobes for localized states. The emission from the S1 mode is distributed over the centres of the Brillouin zones surrounding the central one (marked by yellow hexagons), where the Brillouin zones of a homogeneous lattice are defined. For states S2 and S3, a lobe can spread over two pillars in real space (with a constant phase), and therefore its Fourier image spreads over several Brillouin zones (also with a constant phase).

## 4.2   Analytical solution based on the envelope function approximation

An approximate analytical solution for states S1-S3 can be found by using the envelope function approximation: the states in a modulated lattice can be decomposed into the product $\psi = \psi_B \psi_{env}$, where $\psi_B$ is the eigenstate of the unperturbed-lattice wavefunctions (Bloch wave), and $\psi_{env}$ are the eigenstates of the local modulation potential (envelope function). The gap states emerge from the antisymmetric state located at the top of the first band ($\Gamma$ point of the $\pi^*$ band of graphene). The Bloch function of this state changes its phase by $\pi$ between neighbouring pillars (it goes from positive to negative amplitude in adjacent pillars). If we consider only the $x, y$ coordinates corresponding to the centre of each micropillar, it can be written as

$$\psi_B(x,y) = \sin\left(\frac{\sqrt{3}\pi x}{d} - \frac{\pi y}{d}\right) + \sin\left(\frac{\sqrt{3}\pi x}{d} + \frac{\pi y}{d}\right) - \sin\left(\frac{2\pi y}{d}\right), \tag{3}$$

where $d = 3a/2$, and $a = 2.4$ $\mu$m is the distance between neighbouring sites. On a single hexagon, this antisymmetric function can be written as $\psi_B = \cos 3\varphi$, where $\varphi$ is the polar

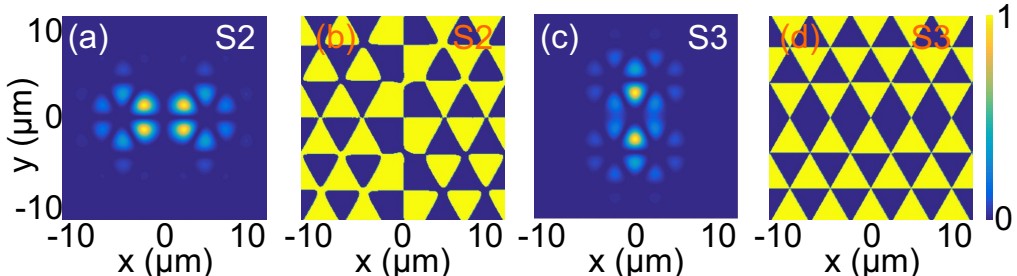

Figure 7: Analytically calculated real space intensity distribution (a),(c) and phase (b),(d) for the states S1 (a)-(b) and S3 (c)-(d). The phase is displayed in units of $\pi$.

angle (choosing the centre of the hexagon as the reference for the polar coordinates). This angular representation is useful to get a qualitative explanation for the separatrix angles of the observed states.

The envelope function $\psi_{env}$ is a solution of the Schrödinger equation for a 2D circular potential trap, effectively created by the reservoir. The solutions in such a trap are characterized by two quantum numbers $n$ (radial) and $l$ (azimuthal). In our case, states S1-S3 all belong to the lowest radial state $n = 0$ and differ only by the azimuthal number (angular momentum). For a finite-size 2D trap of radius $R$, the solution reads [45]

$$\psi_{env}(r,\varphi) = \begin{cases} \frac{e^{il\varphi}}{\sqrt{2\pi}} c_1 J_{|l|}(kr), & k = \sqrt{2|m^*|E/\hbar^2}, \ r \leqslant R, \\ \frac{e^{il\varphi}}{\sqrt{2\pi}} c_2 K_{|l|}(\kappa r), & \kappa = \sqrt{2|m^*|\left(E_g - E\right)/\hbar^2}, \ r > R, \end{cases} \tag{4}$$

where $J$ is the Bessel function of the first kind, and $K$ is the modified Bessel function of the second kind. $c_1$ and $c_2$ are normalisation constants, and $E_g$ is the depth of the trap.

Let us first discuss the angular dependence of $\psi_{env}$. Indeed, it is the angular momentum which defines the energy of the eigenstate of the 2D trap potential considered here. The gap states thus appear in multiplets labelled by the angular momentum of the envelope wavefunction ($l = 0, \pm 1, \pm 2 \ldots$). Since we are dealing with negative mass particles, the highest energy state is the $l = 0$ (this would be the ground state for a circular trap potential for positive mass particles), which is of constant amplitude and phase in the central hexagon ($\psi_{env} = const$). Therefore, the real-space angular distribution of the total wavefunction $\psi_{S1}$ within the central hexagon is determined purely by the Bloch wavefunction $\psi_{S1} = \psi_B = \cos 3\varphi$.

Because of the ellipticity mentioned above, the envelope wavefunctions of states S2 and S3 involve linear combinations of $l = \pm 1$ modes: $\psi_{env,S2} = \chi(r)\sin\varphi$ and $\psi_{env,S3} = \chi(r)\cos\varphi$, respectively, where $\chi(r)$ contains the radial dependence shown in Eq. 4. The full eigenfunctions $\psi_{S2}$ and $\psi_{S3}$ arise from the product of these envelope functions with the Bloch wave, resulting in four distorted lobes in the central hexagon, as we discuss in details below. Outside of the central hexagon, the modulation linked with the envelope function becomes less important, and only the Bloch function (antisymmetric, with six lobes) becomes visible. Other multiplets (e.g. with $l = \pm 2$ envelope function) are also present in numerical simulations, but in the experiment their population is too weak compared to the lasing modes S1-S3. In the central hexagon, the zeros of the state $\psi_{S3} = \psi_B \psi_{env,S3} = \chi(r)\cos 3\varphi \cos\varphi$, corresponding to the change of the sign of the wavefunction, are at 120° and at 60°, whereas those of $\psi_{S2} = \chi(r)\cos 3\varphi \sin\varphi$ are at 90°, which explains the separatrix angles observed in experiment and in numerical simulations.

Finally, we use the full analytical expression for the wavefunction $\psi = \psi_B \psi_{env}$ given by Eqs. (3),(4) to calculate the spatial intensity and phase distributions for the states S2 and S3. To simplify the calculations, we use the asymptotic development of the Bessel functions $J_{|l|}(r) \sim r^{|l|}$ and $K_{|l|}(r) \sim \exp(-r)$ and a convolution with a Gaussian function. The

results are shown in Fig. 7(a,b) for state S2 and (c,d) for S3. The zero-amplitude lines of the wavefunctions appear clearly in the phase plots, at the boundary between regions of different phase. They confirm our analytical predictions for the separatrix angles and agree well with the numerical simulations (compare with Fig. 5(c,f)). The shape of the intensity patterns also matches the experiment (compare with Fig. 4(c,d)): the state S2 is elongated horizontally and S3 is elongated vertically.

It is interesting to compare the profile of the gap states on the central hexagon with the quantized states of an isolated photonic benzene molecule, which can be labelled via the orbital angular momentum $L = 0, \pm 1, \pm 2, 3$ [44]. The antisymmetric intensity and phase profile of gap state S1 in the central hexagon is qualitatively similar to the $L = 3$ wavefunction of the benzene molecule, which also shows six lobes and phase jumps of $\pi$ between adjacent micropillars. However, their origin is different: in the S1 state, the antisymmetric structure arises from the underlying Bloch state, of negative mass, combined with an $l = 0$ envelope state. Gap states S2 and S3 have four lobes, similarly to the linear combinations of $L = \pm 2$ degenerate states in the isolated benzene molecule. The S2 and S3 gap states arise from the interference of $\psi_B = \cos 3\varphi$ and $\psi_{env} = \chi(r) \cos \varphi$ and $\psi_{env} = \chi(r) \sin \varphi$, with separatrix involving $60°$, $90°$ and $120°$, whereas in an isolated benzene molecule the zero-amplitude lines of the states $L = \pm 2$ are given simply by the zeros of $\cos 2\varphi$ and $\sin 2\varphi$, and appear in perpendicular directions ($90°$). The origin of this difference is the interplay between the Bloch wave of the lattice and the envelope function of the asymmetric trapping potential, absent in benzene.

# 5 Gap states lifetime

To explain why lasing occurs in gap states S1-S3 and not in the other possible gap states or in the bulk states, we need to take into account both the lifetime of each state and the overlap of the corresponding wavefunction with the reservoir [7, 12, 46]. The interplay of these two elements establishes which mode reaches the lasing threshold at the lowest excitation density. All the bulk states, extended over the whole lattice, have a weak overlap with the localised reservoir. For a lattice of $30 \times 30$ sites, the overlap of the bulk modes with the reservoir localised in a hexagon is of the order of $I_{bulk,R} \sim w^2/S \sim 10^{-3}$, where $S$ is the patterned area of the cavity. This weak overlap makes lasing in these states very unlikely under localized nonresonant pumping. On the other hand, the overlap of the reservoir with the localized states close to the middle of the gap is roughly of the order of $I_{gap,R} \sim w^2/S_{cell} \sim 0.5$, quite similar for all the gap states. Lasing should therefore preferably occur in gap states versus bulk modes. The gap states showing the highest overlap with the reservoir are those with higher energy (deeper in the gap), showing higher confinement.

The other parameter determining the lasing threshold is the lifetime of each state. The decay of any polariton state in patterned cavities has two contributions: $\Gamma_i = \Gamma_0 + \Gamma_{ext,i}$. The escape of photons through the mirrors is given by $\Gamma_0$, and it is very similar for states whose energy span is much smaller than the Rabi splitting. Based on measurements in an unetched planar microcavity with the same layer structure, we estimate $\Gamma_0$ to be on the order of $1/30$ ps$^{-1}$. The radiative and non-radiative decay of polaritons in the pillar lateral surfaces is described by $\Gamma_{ext,i}$. It depends on the amplitude of the field on the surface of the structure for a given mode, resulting in losses. The field on the surface can be either absorbed or scattered by surface defects. This contribution can be estimated by integrating the amplitude of the field of each state $i$ on the surface of the micropillars (assuming losses are concentrated on the surface itself). Reducing the lattice to a purely two-dimensional system (as in Eq. 1) the field at the

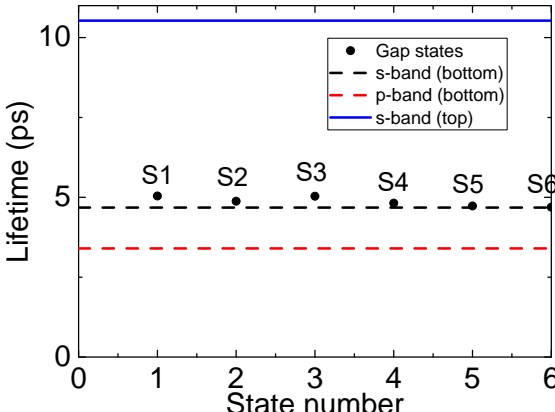

Figure 8: Calculated lifetime of the gap (solid dots) and bulk states (lines).

surface reduces to the field at the external line $d\ell$ defining the etched area:

$$\Gamma_{ext,i} = \alpha \oint_{surf} |\psi_i(x,y)|^2 \, d\ell, \tag{5}$$

where the integration is carried out along the etched surfaces of all the pillars, and $\alpha$ is a constant which accounts for the surface density of losses.

The calculated overall lifetimes of the gap states S1-S6 are shown in Fig. 8, solid dots. We have adjusted the value of $\alpha$ so that the lifetime of the s-band bulk modes is of the same order as that measured in our sample in propagation measurements (not shown). The same value of $\alpha$ is used for the calculation of the lifetime of all states. The gap states with the highest lifetime are S1, S2, and S3, precisely those that lase in the experiment: they are favoured both by their lifetime and by their enhanced overlap with the reservoir. In Fig. 8 we also include the lifetimes of the states located at the bottom of the s- and p-bands (black dashed and red dashed lines), and the top of the s-band (blue solid line). The lifetime of the latter is longer than that of the gap states due to its purely anti-symmetric nature, which confines the mode far from the edges of the micropillar structure. Despite its reduced losses, the very small overlap of this extended mode with the localised reservoir prevents lasing from taking place at the top of the s-band, and favours lasing in the gap modes.

## 6 Conclusions

We have demonstrated polariton lasing in gap states in a photonic honeycomb lattice. The states appear as a consequence of the local potential created by the optically injected polariton reservoir. The shape of the gap states is determined by the product of the anti-symmetric Bloch wavefunction characteristic of the top s-band states and the envelope eigenstates of the two-dimensional quasi-circular potential created by the reservoir. We have shown that a weak deviation from cylindrical symmetry in the reservoir potential reduces the symmetry of the spatial wavefunctions.

The experimental results presented here demonstrate an efficient method to create localised gap states in a lattice of photonic resonators without any need for further processing of the sample. They provide insights into the expected symmetries of the electronic wavefunctions of defect states in honeycomb-like lattices. These have appeared recently as promising sources of single photon emission in two-dimensional solid-state materials [40,41]. However,

their actual geometry is difficult to control, having been hardly studied experimentally. The engineering of the shape of the pumping spot of our photonic lattice provides an efficient method to simulate more elaborate defect architectures involving, for instance, higher orbital modes or fully asymmetric shapes [47].

**Acknowledgements**   We thank Omar Jamadi for fruitful discussions.

**Funding information**   This work was supported by the ERC grant Honeypol, by the EU-FET Proactive grant AQuS, the French National Research Agency (ANR) project Quantum Fluids of Light (ANR-16-CE30-0021), the ANR program "Investissements d'Avenir" through the IDEX-ISITE initiative 16-IDEX-0001 (CAP 20-25), the Labex CEMPI (ANR-11-LABX-0007) and NanoSaclay (ICQOQS, Grant No. ANR-10-LABX-0035), the French RENATECH network, the CPER Photonics for Society P4S, and the Métropole Européenne de Lille via the project TFlight. D.D.S. acknowledges the support of IUF (Institut Universitaire de France).

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
