# Peer review of "Lasing in optically induced gap states in photonic graphene"

_SciPost Physics, doi:SciPost Phys. 5, 064 (2018)_

## Round 1 · Referee Report · Anonymous (Referee 1) · 2018-7-1

Strengths

1. The paper demonstrates an interesting effect (fig.1) which is the emergence of a localised gap state in a honeycomb lattice of coupled micropillars

Weaknesses

1. The emergence of the localised gap state is an interesting and plausible interpretation of the data; nevertheless, focussing 10's of mW onto such a small microstructure inevitably leads to heating and the generation of free carriers. I note the authors have excluded thermo-optic effects by modulating with 1e-6 duty cycle, but how about simple electro-optic effects?
2. The authors claim lasing in the title and throughout the paper. Unfortunatley, the evidence for lasing is very poor. There is still a lot of debate in the community of how to define lasing in such microlasers; the Nature Photonics list is a good start, but the real evidence would be to demonstrate coherence, e.g. via a g2 measurement. In the absence of such a measurement, the minimum they would need to show is a reduction in linewidth by a factor 2 as stipulated by the Shawlow-Townes equation.
3. The authors motivate the paper by stating that the study of 2D materials via "photonic lattice simulators can provide insights in their properties". It would be nice if these insights were better articulated in the conclusion. Instead, they refer to vortex solitons, and it is not particularly clear why these are interesting.

Report

The explanations offered are plausible but need to be better supported by data (cf lasing claim) and the exclusion of conventional effects. Given that there have been many reports of laser, or laser-like action, in a large variety of III-V microstructures, the authors also need to articulate the significance of their findings better.

Requested changes

If the authors can address my three points convincingly, the paper could be considered for publication.

---

## Round 2 · Author Response

We thank the referee for her/his valuable comments on our work. She/he mentions three points that need to be clarified. Here is our answer:
1.- About the possible effects of heating and electro-optic effects.
Thermo-optic effects caused by heating of the semiconductor could indeed lead to a local modification of the optical potential, which could explain the emergence of localised states. As mentioned by the referee, heating is avoided here by modulating the excitation beam with a low duty cycle. When revising the manuscript, we realised that the actual duty cycle is 8x10^-4 instead of 1e-6, with a repetition rate of 0.8kHz. We have corrected this point in the revised version. We have checked that the experimental results do not change when modifying the duty cycle around the value used in the experiment. In any case, any thermo-optic effect would result in a redshift of the local potential at high power, while we observe a blueshift. These arguments allow to safely discard thermal effects.
As for electro-optic effects, they are expected to be extremely weak compared to the modification of the local exciton potential induced by the generation of free carriers and excitons in the quantum wells by the excitation beam at 740 nm. A recent study shows that in microcavities, electro-optic effects may appear in the form of Stark shifts of the energy of the quantum well [A. Hayat et al., PRL 109, 33605 (2012)]. In that study it was shown that extremely high peak powers, only available in pulsed experiments, are needed to induce a blueshift of the order of what we observe: they report 250fs pulses of 2mJ/cm^2 to induce a blueshift of 0.5meV. This corresponds to an equivalent average cw power of 1000W for the spot we use in our experiments. We can thus safely disregard this effect. In the revised version we have added the sentence: “Electro-optics effects can be discarded to be at the origin of the local blueshift due to the very low cw powers used in our experiments compared to the high peak energies required to induce a blueshift via Stark effect [A. Hayat et al., PRL 109, 33605 (2012)].”
2.- About the evidence for lasing.
Our claim of lasing in gap states is supported by two features: (i) the observation of a nonlinear threshold in the emitted intensity as a function of the excitation density (Fig. 1e and Fig. 3g), and (ii) the linewidth reduction across the threshold, which attests the emergence of long temporal coherence. Following the referee’s suggestion, we have now included a detailed analysis of the linewidth as a function of the excitation power for the gap mode in Fig 1 and for the S1 gap mode in Fig. 3. A reduction of the linewidth of a factor of two is observed at threshold. At higher power, nonlinear effects such as spectral wandering due to fluctuating trapped charges degrade the linewidth. We have added a description of the linewidth behaviour as a function of power in the revised text.
3.- About the connection between photonic lattice simulators and the understanding of 2D materials.
We have modified the conclusion section of the manuscript to clarify the interest and potentialities of our study for the understanding and experimental exploration of defect states in two-dimensional lattices.
1.- About the possible effects of heating and electro-optic effects.
Thermo-optic effects caused by heating of the semiconductor could indeed lead to a local modification of the optical potential, which could explain the emergence of localised states. As mentioned by the referee, heating is avoided here by modulating the excitation beam with a low duty cycle. When revising the manuscript, we realised that the actual duty cycle is 8x10^-4 instead of 1e-6, with a repetition rate of 0.8kHz. We have corrected this point in the revised version. We have checked that the experimental results do not change when modifying the duty cycle around the value used in the experiment. In any case, any thermo-optic effect would result in a redshift of the local potential at high power, while we observe a blueshift. These arguments allow to safely discard thermal effects.
As for electro-optic effects, they are expected to be extremely weak compared to the modification of the local exciton potential induced by the generation of free carriers and excitons in the quantum wells by the excitation beam at 740 nm. A recent study shows that in microcavities, electro-optic effects may appear in the form of Stark shifts of the energy of the quantum well [A. Hayat et al., PRL 109, 33605 (2012)]. In that study it was shown that extremely high peak powers, only available in pulsed experiments, are needed to induce a blueshift of the order of what we observe: they report 250fs pulses of 2mJ/cm^2 to induce a blueshift of 0.5meV. This corresponds to an equivalent average cw power of 1000W for the spot we use in our experiments. We can thus safely disregard this effect. In the revised version we have added the sentence: “Electro-optics effects can be discarded to be at the origin of the local blueshift due to the very low cw powers used in our experiments compared to the high peak energies required to induce a blueshift via Stark effect [A. Hayat et al., PRL 109, 33605 (2012)].”
2.- About the evidence for lasing.
Our claim of lasing in gap states is supported by two features: (i) the observation of a nonlinear threshold in the emitted intensity as a function of the excitation density (Fig. 1e and Fig. 3g), and (ii) the linewidth reduction across the threshold, which attests the emergence of long temporal coherence. Following the referee’s suggestion, we have now included a detailed analysis of the linewidth as a function of the excitation power for the gap mode in Fig 1 and for the S1 gap mode in Fig. 3. A reduction of the linewidth of a factor of two is observed at threshold. At higher power, nonlinear effects such as spectral wandering due to fluctuating trapped charges degrade the linewidth. We have added a description of the linewidth behaviour as a function of power in the revised text.
3.- About the connection between photonic lattice simulators and the understanding of 2D materials.
We have modified the conclusion section of the manuscript to clarify the interest and potentialities of our study for the understanding and experimental exploration of defect states in two-dimensional lattices.

---

## Round 2 · List of Changes

- Changed the last sentence of the abstract (page 1)..
- Added a panel to Figs. 1 and 3 showing the linewidth of the lasing gap states.
- Corrected the description of the duty cycle used in the experiments (page 4).
- Added a description of the linewidth narrowing across the threshold (pages 4 and 6).
- Added a discussion about possible electro-optics effects (page 4).
- Modified the conclusion to clarify the connection between our results and the physics of 2D-materials (page 13).

---

## Editorial Decision

published